# Comparison between Peritumoral and Intratumoral Budding in Colorectal Cancer

**DOI:** 10.3390/biomedicines12010212

**Published:** 2024-01-17

**Authors:** Jung-Soo Pyo, Ji Eun Choi, Nae Yu Kim, Kyueng-Whan Min, Dong-Wook Kang

**Affiliations:** 1Department of Pathology, Uijeongbu Eulji Medical Center, Eulji University School of Medicine, Uijeongbu-si 11759, Republic of Korea; jspyo@eulji.ac.kr (J.-S.P.); kyueng@eulji.ac.kr (K.-W.M.); 2Department of Pathology, Chungnam National University Sejong Hospital, 20 Bodeum 7-ro, Sejong 30099, Republic of Korea; b612elf@cnuh.co.kr; 3Department of Internal Medicine, Uijeongbu Eulji Medical Center, Eulji University School of Medicine, Uijeongbu-si 11759, Republic of Korea; naeyu46@eulji.ac.kr; 4Department of Pathology, Chungnam National University School of Medicine, 266 Munhwa Street, Daejeon 35015, Republic of Korea

**Keywords:** colorectal cancer, peritumoral budding, intratumoral budding, prognosis

## Abstract

Tumor budding (TB) is classified, based on location, into peritumoral budding (PTB) or intratumoral budding (ITB). This study aimed to evaluate the relationship between PTB and ITB in colorectal cancers (CRCs). PTB and ITB were investigated and subsequently divided into high and low groups. CRCs were divided into three groups: (1) high PTB/ITB, (2) high PTB or ITB, and (3) low PTB/ITB. The clinicopathological and prognostic significances were evaluated according to the three tumor budding (TB) groups. High PTB/ITB and low PTB/ITB were identified in 32 (12.0%) and 135 (50.8%) patients, respectively. A total of 99 patients (37.2%) were found to have high PTB or ITB. TB was significantly correlated with lymphatic and perineural invasion, lymph node metastasis, metastatic lymph node ratio, distant metastasis, and a higher pTNM stage. A significant correlation was found between high PTB and high ITB (*p* = 0.010). The amount of PTB was found to increase significantly with the amount of ITB (*p* < 0.001) in a linear regression test. Patients with high PTB/ITB had worse overall and recurrence-free survival than those with high PTB or ITB. Conversely, patients with low PTB/ITB had better overall and recurrence-free survival rates than those with high PTB or ITB. However, there was no significant difference in overall and recurrence-free survival between patients with high PTB/low ITB and high ITB/low PTB (*p* = 0.336 and *p* = 0.623, respectively). In summary, the presence of TB, regardless of PTB or ITB, was significantly correlated with aggressive tumor behavior and a worse prognosis than the absence of TB. Additionally, the present study demonstrated that it is feasible to stratify the prognosis of patients based on whether they have both PTB and ITB or only one of the two.

## 1. Introduction

The diagnosis and evaluation of malignant tumors, including colorectal cancer (CRC), is primarily based on histology in pathologic examination. It is well known that a key histological feature indicative of epithelial–mesenchymal transition in these tumors is tumor budding (TB) [1,2,3]. Evaluation factors for malignant tumors are typically described and recommended by the American Joint Committee on Cancer (AJCC) and the College of American Pathologists (CAP) [4]. TB is defined as the presence of single cells or small clusters comprising fewer than five tumor cells at the invasive margin of CRCs [2]. TB is classified according to its location as peritumoral budding (PTB) or intratumoral budding (ITB). According to the guidelines, the amount and grade of PTB, evaluated in a hotspot (area = 0.785 mm^2^), are included in the pathological reports for CRCs [2]. The grades of TB are subclassified by the number of tumor buds as low (0–4), intermediate (5–9), or high (≥10). However, variations in staining methods and differences among investigators can lead to discrepancies in TB assessments. The heterogeneity between the grades of TB can influence the clinicopathological significance of PTB and ITB.

Previous studies have introduced evaluation methods, including hematoxylin and eosin (H&E) and pan-cytokeratin immunohistochemistry (IHC) [2,5]. However, H&E, which is widely used due to its effectiveness in revealing histological details, is the preferred staining method as recommended by the International Tumor Budding Consensus Conference (ITBCC) [2]. According to the ITBCC, only the evaluation of PTB, not ITB, is recommended [2]. The exclusion of ITB from certain evaluations is attributed to the low level of evidence supporting its significance [2]. However, ITB is significantly correlated with lymph node metastasis [6,7], and patients with high ITB generally have a worse prognosis than those with low ITB. This correlation is particularly evident in small intestine adenocarcinoma, where both PTB and ITB are significantly associated with lymph node metastasis and a worse prognosis [1]. In contrast, no definitive correlation has been reported between PTB and the prognosis of other cancers, such as ovarian clear cell carcinoma, as no significant difference in survival has been observed between high and low PTB [8].

Although PTB is linked with aggressive tumor behavior, its correlation with overall survival remains unclear [9]. In colorectal cancer, a significant difference in overall survival has been noted between high and low grades of ITB [9]. This suggests that ITB could be a more reliable prognostic factor in CRC. Furthermore, a previous meta-analysis also highlighted the prognostic impact of ITB in CRC patients [10]. Lugli et al. reported the correlation between PTB and ITB in CRCs and their prognostic implications [6]. However, the relationship between PTB and ITB in various malignant tumors remains an area of uncertainty, largely due to variability in research methods or insufficient studies. Additionally, the prognostic implications of heterogeneous PTB and ITB patterns have not yet been fully explored. Thus, the present study aims to evaluate the relationship between PTB and ITB in CRC patients and to stratify the prognoses of these patients based on PTB and ITB patterns. This study could provide valuable insights into the prognostic significance of PTB and ITB in CRC.

## 2. Materials and Methods

### 2.1. Patients and Specimens

A total of 266 patients who had undergone surgical resection of CRCs at the Eulji University Medical Center, Eulji University School of Medicine (Republic of Korea), were analyzed from 1 January 2001 to 31 December 2010. We meticulously reviewed the medical charts, pathological records, and glass slides to obtain information on clinicopathological characteristics. This included details such as the amount of PTB and ITB, age, sex, tumor size, location, differentiation, tumor depth, as well as vascular, lymphatic, and perineural invasion; lymph node metastasis; the metastatic lymph node ratio; distant metastasis; and the pathologic tumor, node, metastasis (pTNM) stages. We categorized the pathological tumor (pT) stages, represented by pT1-2 and pT3-4 groups. Similarly, the pTNM staging was divided into two groups, encompassing the early stage of pTNM I-II and the advanced stage of pTNM III-IV. This protocol was reviewed and approved by the Institutional Review Board of Uijeongbu Eulji University Hospital (approval no. UEMC 2023-09-006).

### 2.2. Evaluation of Peritumoral and Intratumoral Budding

We investigated the amount of TB in a hotspot (area = 0.785 mm^2^). PTB and ITB were evaluated at the invasive front and within the tumor, respectively. PTB and ITB were divided into two groups, high and low, respectively, based on the amount of TB. Thresholds of ten peritumoral buds and five intratumoral buds were used as the criteria for division into the high and low groups, respectively. TB groups of CRCs were divided into three groups according to the grades of PTB and ITB: (1) high PTB/ITB, (2) high PTB or ITB, and (3) low PTB/ITB. The evaluation of TB was performed by two independent pathologists and discrepancies were resolved through consensus.

### 2.3. Statistical Analysis

Statistical analyses were performed using the SPSS version 22.0 software (SPSS, Chicago, IL, USA). The significance of the relationship between TB and clinicopathological characteristics was determined using either the χ^2^ test or Fisher’s exact test (two-sided). Comparisons between TB and variables such as age, tumor size, and metastatic lymph node ratio were analyzed using a two-tailed Student’s *t*-test. A linear regression test was performed to evaluate whether PTB can be predicted when only ITB is measurable, such as in biopsied samples. Survival curves were estimated using the Kaplan–Meier product-limit method, and differences between the survival curves were determined as significant based on the log-rank test. The results were considered statistically significant at *p* < 0.05.

## 3. Results

### 3.1. The Clinicopathological Significances of Peritumoral and Intratumoral Budding

Representative PTB and ITB images are shown in Figure 1. Histologic features of high PTB and ITB are characterized by the presence of small clusters or isolated tumor cells detaching from the main tumor and infiltrating the surrounding stroma.

Table 1 shows the relationship between the presence of PTB and ITB and various clinicopathological parameters in CRCs. CRCs with high PTB and ITB comprised 32 of 266 CRCs (12.0%). Among the 266 patients, 99 (37.2%) had high PTB or ITB. Low PTB and ITB were identified in 135 of the 266 patients (50.8%). Patients with low PTB and ITB showed significantly lower lymphatic invasion, perineural invasion, lymph node metastasis, metastatic lymph node ratio, distant metastasis, and pTNM stage compared to those with high PTB and/or high ITB. This suggests that lower levels of PTB and ITB are associated with less aggressive tumor characteristics and possibly a better prognosis. Additionally, the high PTB/ITB group showed a greater prevalence of advanced pTNM stages, indicating a potential correlation between high PTB/ITB and more advanced disease stages in CRCs. However, there were no significant differences in sex, tumor size, location, tumor differentiation, vascular invasion, or pT stage between the TB groups.

### 3.2. The Comparison between Peritumoral and Intratumoral Budding

Next, the correlation between PTB and ITB was investigated. High PTB was significantly correlated with high ITB (*p* = 0.010; Table 2). High ITB was found in 27 of the 162 patients with low PTB (16.7%). Among patients with high ITB, the high PTB rate was 54.2% (32/59). The amount of PTB significantly increased alongside the amount of ITB (*p* < 0.001) in a linear regression test (Figure 2).

### 3.3. Prognostic Stratification Using Peritumoral and Intratumoral Buds

The prognostic stratification of the three TB groups was investigated. Patients with high PTB/ITB had worse overall and recurrence-free survival than those with high PTB or ITB (Figure 3). Patients with low PTB/ITB had better overall and recurrence-free survival rates than those with high PTB or ITB. However, there was no significant difference in overall and recurrence-free survival between patients with high PTB/low ITB and high ITB/low PTB (*p* = 0.336 and *p* = 0.623, respectively; Appendix A).

## 4. Discussion

According to the ITBCC, TB is defined as a single tumor cell or a small cluster of less than five cells at the invasive margin of CRCs [2]. TB is evaluated at the invasive front, but not within the tumor in the ITBCC’s methods [2]. Only PTB, not ITB, is recommended for evaluation in CRCs [2]. The predictive roles of PTB in CRCs have been previously reported. Additionally, since TB is evaluated using routine H&E staining, this method is easily applicable. However, in some cases, the evaluation of TB is possible using biopsy specimens before obtaining surgical specimens. In this case, ITB—in which TB is present within the tumor—is appropriate for the evaluation of TB from biopsy specimens. Previous studies have reported a relationship between PTB and ITB in CRCs [6,11]. However, detailed information on the prognosis was not available from previous studies. In our research, we analyzed in detail the correlation between the number of buds and the grades of PTB and ITB. Additionally, the prognoses of patients with CRC were stratified by grouping PTB and ITB. To our knowledge, this study is the first attempt to evaluate prognostic stratification using PTB and ITB in CRCs. The results of our study are as follows: (1) CRCs with high PTB and ITB showed more aggressive tumor behavior and worse prognoses than the remaining CRCs. (2) The ITB pattern can further stratify the prognosis in CRC with low PTB. (3) The amount of ITB significantly increased in correlation with the amount of PTB. These findings underscore the importance of both PTB and ITB in understanding and predicting the progression of CRCs. Further research could explore the integration of PTB and ITB evaluations with advanced diagnostic tools, such as whole slide imaging for primary pathology diagnosis, for a more comprehensive understanding of CRCs.

As described above, TB is investigated at invasive fronts and intratumoral areas and is classified as PTB or ITB, respectively. PTB is evaluated at the invasive front, a region that can be easily identified. However, this approach may not fully represent all tumor growth and proliferative abilities, as not all tumor growth is directed toward the invasive front. Therefore, evaluating the differences between high and low PTB could lead to more detailed tumor classification, enhancing the prediction of tumor behavior and patient prognosis. Our study compared the number of tumor buds between PTB and ITB groups, finding that the rate of high ITB was significantly higher in CRCs with high PTB compared to those with low PTB. This suggests that PTB evaluation alone might not cover the complete scope of tumor growth characteristics. Future studies could focus on refining the criteria for TB assessment, potentially incorporating quantitative measures and digital pathology for more accurate and reproducible evaluations. The present study further investigated the prognoses of patients with high PTB and ITB. It was found that 12.0% of all cases had high PTB and ITB, with 27.1% having high PTB and 10.2% having high ITB. Interestingly, CRCs with low PTB and low ITB, which were associated with a better prognosis, were observed in 50.8% of the patients. We explored the following two questions in more detail. First, what is the prognosis of patients with high PTB and ITB? The second question was whether there was a prognostic difference between CRCs with PTB or ITB only. Our study indicated that CRC patients with both high PTB and ITB had a worse prognosis compared to other groups. The prognosis was generally worse in CRC patients with TB, regardless of PTB or ITB status. However, no significant difference in survival was noted between groups with only high PTB or ITB. This highlights the importance of considering both PTB and ITB in evaluating CRC prognosis. Incorporating both PTB and ITB in prognostic assessments could lead to more effective patient management and treatment planning.

Giger et al. compared the ITB of biopsy specimens and the PTB of surgical specimens [11]. They reported a significant correlation between ITB and PTB [11]. The concordance rate between high ITB and high PTB cases was as high as 91.7% (11 of 12 cases) [11]. However, in cases with low ITB, only 50% showed concordance with low PTB. These results suggest that it may be difficult to predict PTB status in cases with low ITB. Notably, their study did not evaluate the correlation between ITB and PTB in the surgical specimens. In our study, the positive and negative predictive values of ITB for high and low PTB were 54.2% and 65.2%, respectively. These values provide a quantitative measure of ITB’s predictive accuracy for PTB status. Lugli et al. reported a significant correlation between ITB and PTB through a linear regression test [6]. Similarly, they found that the number of PTB cases significantly increased with the amount of ITB, echoing our results.

Interestingly, a previous study measured TB using H&E and pan-cytokeratin IHC, investigating the correlation between PTB and ITB in CRCs [5]. The correlations between PTB and ITB differed according to the staining method used (H&E staining or pan-cytokeratin IHC). Statistical significance was identified with H&E staining, but not with pan-cytokeratin IHC [5]. This indicates that H&E staining may be more reliable for detecting correlations between PTB and ITB. The concordance rates of PTB and ITB between H&E and pan-cytokeratin were 42% and 30%, respectively [5]. Different staining methods and observers may cause discrepancies. However, a low concordance rate indicates a need for more precise criteria, which requires further research.

In small intestine adenocarcinomas, the positive predictive value and negative predictive value of ITB for PTB were 98.4% and 56.1%, respectively [1]. PTB and ITB are significantly correlated with lymph node metastasis and a worse prognosis in small intestine adenocarcinomas [1]. This correlation underscores the importance of both PTB and ITB in the prognostic assessment of these tumors. In esophageal adenocarcinomas, PTB and ITB were significantly correlated with aggressive tumor behavior [9]. However, there was a significant correlation between prognosis and ITB, but not PTB [9]. There was a significant overall survival difference between high and low grades of ITB [8,9]. In ovarian clear cell carcinoma, no significant difference in survival between high and low PTB was identified [8]. However, a significant correlation was found between ITB and prognosis [8]. Tumor characteristics and histological patterns can influence the prognostic impact of TB. Therefore, further studies are needed to confirm the clinicopathological significance of PTB and ITB based on the characteristics of each tumor.

In cases where evaluating PTB is challenging, especially in preoperative rectal cancers, the evaluation of ITB can serve as a viable alternative. If ITB assessment can approximate PTB, the measurement of both PTB and ITB can complement each other in providing a more comprehensive understanding of tumor characteristics. In the present study, a linear correlation was observed between the amounts of PTB and ITB. Additionally, both PTB and ITB were found to be associated with lymph node metastasis and poor prognosis. Our study aimed to determine whether prognostic stratification could be achieved within the same PTB group. The results indicated that even within the low PTB category, prognostic stratification could be based on whether ITB was high or low. This finding implies that patients with low PTB may exhibit heterogeneity in terms of prognosis. Considering that the ITBCC recommends evaluating PTB alone, this study suggests that PTB evaluation alone might not reliably predict prognosis. In preoperative biopsy specimens, the evaluation of tumor budding primarily relies on ITB, given the absence of an established prognostic model for ITB in these specimens. Furthermore, assessing the correlation of PTB in surgical specimens after concurrent chemoradiation therapy may be challenging, making the study’s findings even more significant. Based on the results, ITB assessment from biopsies can provide valuable information about aggressive tumor behavior, such as lymph node metastasis, especially when ITB is high. Subsequently, during surgery, PTB measurement can be performed alongside the existing or newly obtained ITB assessment to make a more detailed prediction of the patient’s prognosis. This comprehensive approach could lead to more personalized and effective treatment strategies, ultimately improving patient outcomes. If the biopsy specimen indicates low ITB, it is advisable to confirm this finding by double-checking the surgical specimen. This practice could help identify cases where biopsy results may not fully represent the tumor’s invasive potential, ensuring more accurate staging and treatment planning.

## 5. Conclusions

TB, whether PTB or ITB, was significantly correlated with aggressive tumor behavior, lymphatic and perineural invasion, lymph node metastasis, distant metastasis, and a high pTNM stage. Patients with high TB had a worse prognosis than those with low TB. In addition, it was feasible to stratify the prognoses of patients based on whether they had both PTB and ITB or only one of the two.

## Figures and Tables

**Figure 1 biomedicines-12-00212-f001:**
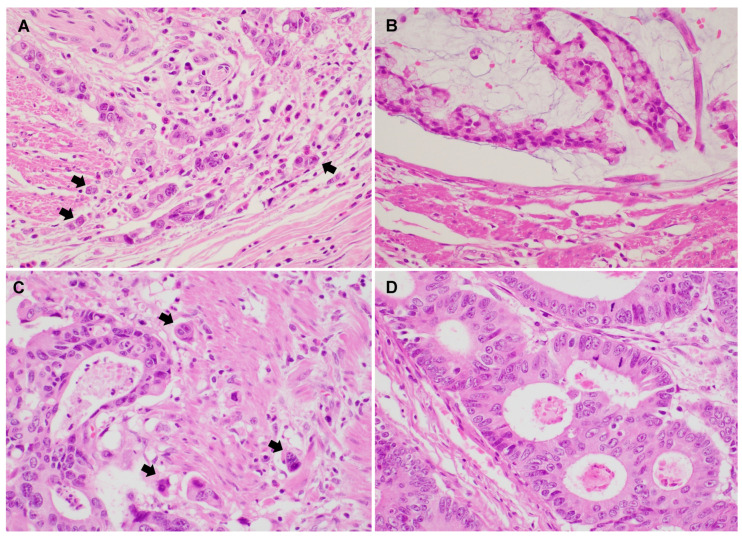
Representative images show colorectal adenocarcinoma with peritumoral budding (PTB) and intratumoral budding (ITB) (**A**–**D**). (**A**) Conventional colorectal adenocarcinoma showing high PTB (×400). (**B**) Conventional colorectal adenocarcinoma showing low PTB (×400). (**C**) Colorectal adenocarcinoma with high ITB (×400). (**D**) Colorectal adenocarcinoma with low ITB (×400). (Arrow: tumor buds).

**Figure 2 biomedicines-12-00212-f002:**
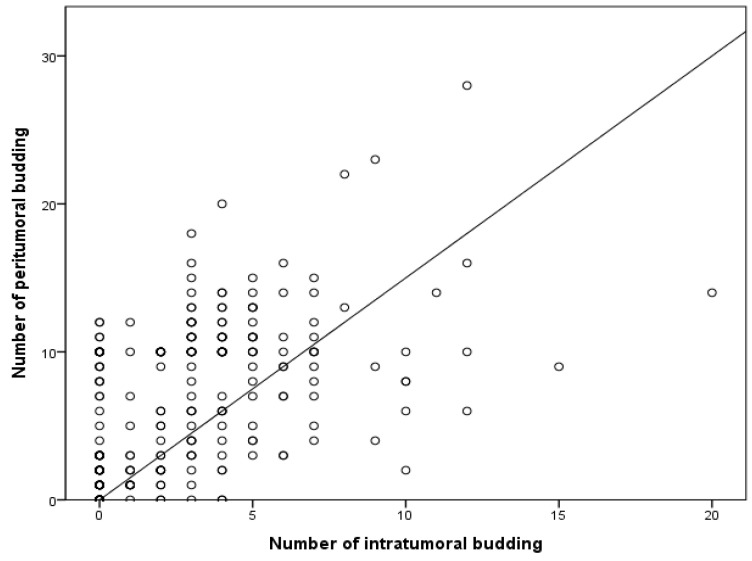
Correlation between the numbers of intratumoral and peritumoral buds based on linear regression. R^2^ = 0.267; *p* < 0.001.

**Figure 3 biomedicines-12-00212-f003:**
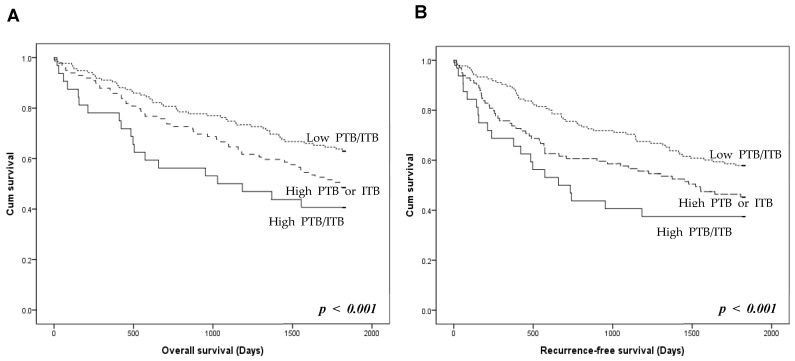
Prognosis according to the presence of peritumoral budding (PTB) and intratumoral budding (ITB) in colorectal cancers. (**A**) Overall survival of the three tumor budding (TB) groups. (**B**) Recurrence-free survival of the TB groups. High PTB/ITB group shows worse overall and recurrence-free survivals than those with high PTB or ITB. Low PTB/ITB group shows better overall and recurrence-free survivals than high PTB or ITB (log-rank, *p* < 0.001). *p* < 0.05 are highlighted in italic bold.

**Table 1 biomedicines-12-00212-t001:** The comparison between the presence of peritumoral budding (PTB) and intratumoral budding (ITB) and clinicopathological parameters in colorectal cancers.

	High PTB/ITB	High PTB or ITB	Low PTB/ITB	*p*-Value	Coefficient
Total (*n* = 266)	32 (12.0)	99 (37.2)	135 (50.8)		
Age (years)	66.00 ± 12.24	62.87 ± 13.11	63.54 ± 12.94	0.492	
Sex					
Male	15 (46.9)	51 (51.5)	69 (51.1)	0.895	0.029
Female	17 (53.1)	48 (48.5)	66 (48.9)		
Tumor size					
≤5 cm	17 (53.1)	33 (33.3)	56 (41.5)	0.119	0.126
>5 cm	15 (46.9)	66 (66.7)	79 (58.5)		
Tumor size (cm)	4.96 ± 1.71	5.64 ± 1.91	5.45 ± 2.26	0.268	
Location of tumor					
Right	17 (53.1)	40 (40.4)	71 (52.6)	0.152	0.119
Left	15 (46.9)	59 (59.6)	64 (47.4)		
Tumor differentiation					
Well or Moderate	22 (68.8)	76 (76.8)	113 (83.7)	0.125	0.125
Poor	10 (31.3)	23 (23.2)	22 (16.3)		
Vascular invasion					
Present	5 (15.6)	11 (11.1)	8 (5.9)	0.149	0.120
Absent	27 (84.4)	88 (88.9)	127 (94.1)		
Lymphatic invasion					
Present	17 (53.1)	31 (31.3)	22 (16.3)	** *<0.001* **	0.275
Absent	15 (46.9)	68 (68.7)	113 (83.7)		
Perineural invasion					
Present	14 (43.8)	16 (16.2)	14 (10.4)	** *<0.001* **	0.280
Absent	18 (56.3)	83 (83.8)	121 (89.6)		
pT stage					
pT1–2	3 (9.4)	12 (12.1)	26 (19.3)	0.197	0.111
pT3–4	29 (90.6)	87 (87.9)	109 (80.7)		
Lymph node metastasis					
Present	22 (68.8)	63 (63.6)	61 (45.2)	** *0.005* **	0.200
Absent	10 (31.3)	36 (36.4)	74 (54.8)		
Metastatic lymph node ratio	0.24 ± 0.28	0.15 ± 0.23	0.10 ± 0.20	** *0.002* **	
Distant metastasis					
Present	9 (28.1)	11 (11.1)	9 (6.7)	** *0.002* **	0.215
Absent	23 (71.9)	88 (88.9)	126 (93.3)		
pTNM stage					
I–II	10 (31.3)	34 (34.3)	71 (52.6)	** *0.007* **	0.193
III–IV	22 (68.8)	65 (65.7)	64 (47.4)		

Numbers in parentheses represent percentages. *p* < 0.05 are highlighted in italic bold.

**Table 2 biomedicines-12-00212-t002:** The comparison between the presence of peritumoral and intratumoral budding in colorectal cancers.

	Peritumoral Budding		*p*-Value	Coefficient
	High	Low		
Total (*n* = 266)	104 (39.1)	162 (60.9)		
Intratumoral budding				
High	32 (30.8)	27 (16.7)		
Low	72 (69.2)	135 (83.3)	** *0.010* **	0.345

Numbers in parentheses represent percentages. *p* < 0.05 are highlighted in italic bold.

## Data Availability

The datasets generated and/or analyzed during the current study are available from the corresponding author on reasonable request.

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
