# Peer review of "Comparison between Peritumoral and Intratumoral Budding in Colorectal Cancer"

_biomedicines, 2024, doi:10.3390/biomedicines12010212_

Round 1
Reviewer 1 Report
Comments and Suggestions for Authors
This is a manuscript of a study that compared peritumoral and intratumoral budding in colorectal cancer. There are several concerns with the study design and methodology that need clarification. Below are comments for author consideration.
1. Title: This must be revised. What is meant by the term "detailed analysis"?
2. Abstract: The aim of the study should be stated using measurable verbs. The word "elucidate" is not a measurable verb. Consult the Bloom's taxonomy for appropriate terms. The word "correlation" is a specific type of statistical analysis. Reporting the results of a correction requires including the magnitude of the estimate, the correlation coefficient. The correlation coefficient should be included in the results along with the p-values. Spell out "TB" in full before the first use.
3. Introduction: There are several errors in this section of the manuscript. A few are mentioned here. What is meant by "daily practice", "daily routine"? In line 41, and other parts of the manuscript, the definition of tumoral budding, TB, is incomplete. Revise this.
4. Methods: The primary method for calculating correlation is not by x2 test and linear regression...Lines 97-100. Explain why this method was used.
6. Results: The correlation coefficients should be included in the results narration and tables 1 and 2. It is important to know the magnitude and direction of the correlation. This is missing in the results.
Discussion: The definition of "TB" is incomplete. Revise this.
See attached file with highlighted text for your convenience during revision.
Best of luck!

There are errors in grammar and choice of words as well as incomplete sentences that should be revised.
Author Response
We tried to address the points raised by the reviewers as best as we could.
Please see the attachment.

Reviewer 2 Report
Comments and Suggestions for Authors
The authors did a good job depicting clinical significance of PTB and ITB. One minor suggestion - since it can be challenging for unqualified experts to distinguish between low and high levels of ITB/PTB, the authors could include a more thorough visual explanation of figure 1.
Author Response

(The authors gave the same response as above.)

Round 2
Reviewer 1 Report
Comments and Suggestions for Authors
The authors have adequately responded to my comments from the initial review.
Comments on the Quality of English LanguageMinor editing required for grammar.